# THz Multi-Mode Q-Plate with a Fixed Rate of Change of the Optical Axis Using Form Birefringence

**DOI:** 10.3390/mi13050796

**Published:** 2022-05-20

**Authors:** Can Koral, Zahra Mazaheri, Antonello Andreone

**Affiliations:** 1Department of Physics, University of Naples Federico II, 80126 Naples, Italy; zahra.mazaheri@unina.it (Z.M.); andreone@unina.it (A.A.); 2Naples Unit, National Institute of Nuclear Physics, 80126 Naples, Italy

**Keywords:** terahertz devices, q-plate, hyperspectral imaging, birefringence, vector beams, 3D printing

## Abstract

We report the design, fabrication and experimental validation of a THz all-dielectric multi-mode q-plate having a fixed rate of change of the optical axis. The device consists of space-variant birefringent slabs manufactured by 3D printing using melt-extruded Acrylonitrile Butadiene Styrene (ABS). The desired form birefringence is analytically evaluated and experimentally measured by the THz time domain spectroscopy technique. The manufactured q-plate design is characterized using a polarization-sensitive imaging setup. The full electric field spatial maps are acquired from the beam propagating through the q-plate. The device enables the realization of both radial and azimuthal vector beams at discrete frequency intervals by controlling the space-dependent orientation of the ordinary and extraordinary axes in the transverse plane with a multi-mode sequence.

## 1. Introduction

With the continuous progress in THz wave generation and detection techniques, technologies featuring high field intensities (ultrastrong electromagnetic field) and high repetition rates are of increasing interest. Combined with major peculiarities such as a high coherency, nonionizing nature and transparency to many dielectric materials, the THz spectrum holds a growing potential for probing and controlling a variety of complex phenomena, ranging from the fundamental processes of molecules and nanostructures to phase transitions in solids and dynamical interactions in biomaterials.

The THz band covers the elementary rotational and vibrational excitations of many materials in nature. Apart from the major use of THz waves to extract information on the electro-optical properties as a standard spectroscopic tool, the sensitive control over the polarization states in novel and complex devices. Such as metamaterials, and/or field-field interactions, such as optical/THz, IR/THz and X-Ray/THz probes in condensed matter systems, is recently becoming a major field of research. Recent advances have shown the potential of using THz beams with tailored profiles achieving a higher degree of control over matter and intense field transients [1]. In this context, beam shaping and controlled manipulation over its amplitude, phase, polarization, and spatial degrees of freedom correlated with the orbital angular momentum play a key role in optics and photonics.

Even if today the available structures in the THz regime are still limited, due to the constraint of the present techniques of light manipulation, THz beams carrying optical angular momentum have been proposed in many applications, ranging from THz communications to super-resolution imaging and the investigation of nonlinearities in advanced materials and systems [2,3,4], and even to accelerate and manipulate electron bunches [5]. For instance, coherent high-field THz radiation from ultrashort relativistic electron bunches is generated by accelerator-based sources, where the super-radiant and the high field emission require density modulation at THz frequencies [6]. In these novel experiments using electron bunch triggered radiation, the THz emission becomes self-synchronized with all radiation types (from IR to X-ray) [7]. Beam modulation and polarization state control become key factors to use coherent THz pulses for electron beam diagnostics and even to extract information on the charge distribution of the electron bunches.

Some recent works proved this concept, achieving the linear acceleration of relativistic electron beams using polarization-tailored THz pulses [8]. The authors designed a THz beam manipulator for phase-velocity matching with the electron bunch. At the design frequency of 0.4 THz, the dispersion relation results in propagation at a super relativistic phase velocity (v_p_ = 0.999c) with a cut-off frequency at f_c_ = 0.23 THz.

Electromagnetic beams with modified phase structures may carry a quantized amount of Orbital Angular Momentum (OAM) [9]. Beams with OAM are attractive for increasing the THz wireless communication capacity [2]. Moreover, recent works have demonstrated the use of OAM beams for the imaging of materials below the diffraction limit, with nonlinearities achieving an ultrahigh spatial resolution [10]. Complex electromagnetic field profiles can be formed under the controlled manipulation of spatial amplitude and/or phase in a point-to-point manner over the plane of incidence of the designed structures, in such a way transforming the optical angular momentum from the spin to the orbital form due to the anisotropy of the dispersion relation in the magnetic field. Specific cases include Bessel, Airy, vector and vortex beams.

Successful demonstrations have been accomplished mainly in the optical regime by using liquid crystals or intrinsically highly birefringent crystals. In these systems, beam front manipulation is realized by using axially symmetric wave plates as the main building blocks [11]. It is worth noting that liquid crystal-based systems might become a weak point for high field terahertz applications, due to the temperature dependence of their anisotropic response. Moreover, at THz frequencies, these commonly used approaches are difficult to apply, because of the high absorbance and/or the intrinsic limitations of the materials, such as wavelengths being far larger with respect to the birefringent media structures. A possible solution to this problem can be realized by using relatively advanced plasmonic structures. Some recent studies have shown the possibility of actively manipulating the spin and orbital angular momentums at THz frequencies by using graphene-based hybrid plasmonic waveguide structures [12,13,14].

Another solution to overcome these obstacles in a novel and effective way is to use periodically structured dielectric interfaces to create high-birefringence media [15,16]. The birefringence associated with these structures is often called “form birefringence” [17] and has been applied to a variety of devices such as wave plates, filters and phase shifters. When the field wavelength is much larger than the dimensions of the individual components, the traversing beam senses the structure in terms of the effective optical parameters. In this way, one can manipulate the system with ease, by using relatively transparent materials such as polymers and even air in a periodic sequence to engineer highly birefringent media. Most importantly, the resulting effective birefringence can be tailored by changing the design parameters accordingly, such as material refractive index and periodicity and thickness of the components involved in the structure.

In the sub-THz window, geometrical limits are on the millimeter scale; therefore. additive manufacturing techniques based on commercially available three dimensional (3D) printers and melt-extruded dielectric filaments hold the potential to quickly and efficiently produce prototypal devices in a cost-effective, environmentally friendly way [18]. Some recent works report the possible use of 3D-printed shadow masks on silicon [19], showing the feasibility of applying lithography and ion etching techniques to create micromachined subsystems gaining control over the polarization states of the electromagnetic field in a relatively easier and faster way.

Polymeric materials highly benefit from low permittivity, absorbance and dispersion in the THz regime [20,21] and are easily available on the market, including their composites containing metallic particulates and even carbon nanoparticles. Using 3D printing, therefore, one can easily produce form birefringent devices based on different kinds of melt-extruded materials and air.

## 2. Materials and Methods

The achievable “form birefringence” for a given air–dielectric groove dual system can be analytically calculated looking at the effective quasi-static refractive index for the electrical field parallel (*n_TE_*_,0_) and perpendicular (*n_TM_*_,0_) with respect to the groove pattern orientation [22]:(1)$$n_{TE,0}=\sqrt{V_{1}+V_{2}n_{2}{}^{2}}\;$$
(2)$$n_{TM,0}=\sqrt{\frac{1}{V_{1}+V_{2}/n_{2}{}^{2}}}$$
where $n_{1}=n_{air}=1$, $n_{2}$ is the refractive index of the polymer, and $V_{1}$ $\mathrm{and}\;V_{2}$ are the volumetric fractions of the air grooves and polymer, respectively. The above equations provide an adequate estimate of the structural birefringence for the given bulk refractive indices and design parameters under an Effective Medium Theory (EMT) approach. However, it is important to observe that this estimation is violated when the periodicity of the structure exceeds the wavelength of the impinging radiation. In such a case, 2nd-order EMT solutions must be used [23]:(3)$$n_{TE}={\left[n_{TE,0}{}^{2}+\frac{1}{3}{\left(\frac{\Lambda }{\lambda }\pi f_{1}f_{2}\left(n_{1}{}^{2}-n_{2}{}^{2}\right)\right)}^{2}\right]}^{1/2}\;$$
(4)$$n_{TM}={\left[n_{TM,0}{}^{2}+\frac{1}{3}{\left(\frac{\Lambda }{\lambda }\pi f_{1}f_{2}\left(\frac{1}{n_{1}{}^{2}}-\frac{1}{n_{2}{}^{2}}\right)n_{TE,0}n_{TM,0}{}^{3}\right)}^{2}\right]}^{1/2}$$
where $\Lambda =L_{1}+L_{2}$ is the period given by the structure and the separation widths *L*_1_ and *L*_2_, for which the volume fractions of the structures with equal thicknesses are given by $f_{1}=\frac{L_{1}}{\left(L_{1}+L_{2}\right)}$ and $f_{2}=1-f_{1}$.

As clearly seen from the above equations, the given periodic structures result in a positive dispersion even for nondispersive materials, and this assumption holds its validity at wavelengths close to the quasi-static limit due to the Λ/λ relation. Moreover, when the periodicity exceeds the impinging wavelength, scattering and wave-guiding effects become dominant at higher frequencies, limiting the operational bandwidth of the device. In the case of devices realized using 3D printing, the cut-off frequency can be estimated in a trivial way by taking into consideration the limits of extrusion features, i.e., the printer nozzle size. Assuming a feature size of 400 µm, for a periodicity of 800 µm, bandwidth devices working up to 400 GHz are feasible.

The devices presented in this work were manufactured by a commercial 3D printer (3D DREMEL^®^ DigiLab 3D45) by melt extruding Acrylonitrile Butadiene Styrene (ABS) filaments (ECO-ABS (ECO-BLA-01)). Bulk ABS disks with various thicknesses were printed to investigate the index of refraction of the material by using a standard THz time domain spectrometer system (Menlo Systems TERA K-15) in transmission mode. The frequency-dependent complex response was extracted by means of a commercial software that uses a total variation technique to iteratively minimize the periodic Fabry−Perot oscillations produced inside the material [24]. ABS shows an index of refraction n_ABS_ = 1.68 with a flat frequency response in the spectral region of interest.

Using 3D printing, a THz *λ*/4 waveplate based on a form birefringence design was realized by alternating layers of ABS and air with equidistant interface separations of 400 µm. The main advantage of using equidistant interface separations ($f_{1}=\frac{L_{1}}{\left(L_{1}+L_{2}\right)}$ = 0.5) is that the birefringence variation on the structure thickness during manufacturing is minimized, as shown in Figure 1a. This also allows us to attain the radial and azimuthal beam vectors at the exact design frequency. We measured the THz E-Field intensity maxima acquired by rotating the manufactured *λ*/4 plate along the THz beam propagation axis with 15° steps, as shown in Figure 1b. For comparison, the transmittance of an ideal quarter wave plate, given by $1-\left(1/2\right)\sin^{2}\left(2\phi \right)$, is also plotted in the same graph using a black continuous line.

In Figure 2, the measured effective refractive indices for the TM and TE configurations are compared with the EMT static and the 2nd-order solutions (long-dashed and short-dashed curves, respectively). Following the definition, TE (TM) refers to the THz transmission when the grid structure is parallel (perpendicular) with respect to the E-Field polarization direction.

The birefringence calculated from both solutions is similar and well explains the experimental results. However, static solutions (Equations (1) and (2)) contain no information on the cut-off frequency since the extracted index of refraction is constant. Second-order EMT solutions (Equations (3) and (4)) give instead an estimate of the cutoff limit of the designed birefringent system as the extracted index of refraction values reaches and exceeds the values of the ABS material. This behavior is more clearly observed with the experimentally extracted parameters where they deviate from the calculated values after around 0.3 THz (Figure 2).

## 3. Q-Plate Design and Characterization

A q-plate is a pure geometrical phase optical element enabling the realization of vector and vortex beams based on the diverse space-dependent orientation of the ordinary and extraordinary axes in the transverse plane by using a birefringent waveplate.

The most commonly used form in the THz regime was adapted from the radially variant q design, the so called meta q-plate [25]. In this design, the changing rate of the optical axis with respect to the azimuthal angle (q is usually an integer or a semi-integer number) manipulates the beam front in a specifically engineered geometry. This technique allows generating inhomogeneous beams containing polarization singularities and/or specific phases. Similar geometries were later adapted and manufactured using 3D printing [26].

Differently, using the form birefringence, we created a q-plate with a fixed rate of change of the optical axis with respect to the azimuthal angle. Also studied in the literature, the rotationally symmetric plate design is expected to give rise to perfect spin-to-orbital angular momentum conversion, with no angular momentum transfer to the plate [27].

This design is based on the implementation of the previously characterized *λ*/4 waveplate in a segmented geometry having space-variant birefringent slabs with an azimuthally varying in-plane optical axis orientation of 22.5°. The manufactured device is shown in Figure 3a. When the incident field is oriented along the fast axis of the q-plate, it produces a radial polarization distribution, corresponding to the polarization direction of the impinging E-field (blue arrow) parallel to the fast axis (red arrow) of the q-plate (see Figure 3b). Similarly, when the incident field is oriented perpendicularly to the main axis of the device, it produces an azimuthal polarization distribution, corresponding to the polarization direction of the impinging E-field perpendicular to the fast axis of the q-plate (see Figure 3c).

The design frequency and the device thickness of a waveplate can be estimated from the expected phase retardation (Γ) of a multi-order wave plate:(5)$$\frac{2\pi }{\lambda_{design}}d_{design}\Delta n=2m\pi +\Gamma$$
where Γ=π for a *λ*/2 waveplate and $\Gamma =\frac{\pi }{2}\;$for a *λ*/4 waveplate, and m is an integer. Following this equation, the polarization component along the slow axis delays with an order of m. The 4 mm device thickness for the experimentally measured birefringence Δn=0.14 corresponds to a design frequency of 133 GHz for the *λ*/4 waveplate, at the zero order (m = 0).

In the THz region, the birefringence achieved using periodically structured dielectric/air interfaces are comparable with levels attained by commercially available liquid crystals [28]. By replacing the polymer with silicon would be an alternative to increase the performance of the q-plate, and also with the advent of new technologies in the 3D printing materials/filaments such as carbon/graphene/MWCNT filler-based filaments [21,29], it is expected to reach higher levels of birefringence.

## 4. Measurements and Discussion

The full position-dependent electric field of the THz beam propagating through the q-plate was obtained by a standard raster scan imaging routine. A standard transmission geometry with four TPX lenses was used, producing a beam spot of approximately 5 mm, much larger than the dielectric slab width (400 μm). The device was placed on a computer controlled two-axis scanning unit, covering a 3 × 3 cm^2^ surface with a 0.5 mm step. A THz time domain signal within 120 ps interval was acquired for each step.

The x-component *E_x_* was acquired having the emitter and detector PCA polarizations in the same direction, whereas the y-component *E_y_* was obtained having the detector rotated by 90°. The full E-field response of the designed q-plate with respect to the fast axis was obtained by rotating the device with 90° steps and repeating each measurement. As the photoconductive detector is only sensitive to the electric field in one linear polarization direction, it allowed us to monitor both the intensity and phase changes from the general shape of the time domain signal, since the peak intensity is directly proportional to the E-field strength, whereas the relative direction (up (+) or down (−)) of the first maximum gives information on the E-field direction vector (±). A previous study has successfully reconstructed the spatial distribution of a THz vortex beam to depict the E-Field intensity and the related phase changes in a vector notation from the THz time domain signals [30]. Using a similar approach, we mapped the peak position of the THz waveforms shown in Figure 4. The given images show how the electromagnetic field is being manipulated by using the given q-plate structure. As is clearly seen in the E-Field maps, the x- and y-electric field components interchange themselves, showing that the polarization state switches from a radial to azimuthal orientation. This observation fully agrees with previously reported findings on the functionality of diverse q-plate geometries [26,30,31,32].

It is important to note that the in-plane optical axis orientation 22.5° (π/8) gives a consecutive retardation in between each segment of the device, which corresponds to a discrete shift of 67 GHz starting from the zero-order frequency. Therefore, the device is designed to show the expected E-Field response in a multi-mode sequence ($f_{design}+m*67$) [GHz] at the frequencies listed in Table 1. The frequencies corresponding to the *λ*/2 waveplate are also tabulated to underline the coinciding frequency (268 GHz for the corresponding zero order) with the *λ*/4 waveplate.

This allowed us to attain the radial and azimuthal vector beams at discrete frequency intervals starting from the zero-order design frequency of the device, by controlling the space-dependent orientation of the ordinary and extraordinary axes in the transverse plane with a multi-mode sequence. As previously shown in Figure 2, the form birefringence loses its linearity as the frequency exceeds 300 GHz. For this reason, the calculations must be elaborated taking into account this limitation. In Table 1, calculations carried out using Equation (5) for d_design_ = 4 mm and birefringence Δn=0.14 are tabulated. The coinciding *λ*/4 and the *λ*/2 waveplate frequencies are shown with an asterisk. The multi-mode frequencies where the azimuthal to radial conversion was observed are tabled in bold.

The information on the field vectors was retrieved by combining the amplitude and the phase extracted from the FFT analysis. The corresponding vector fields $(E_{Y}exp^{-i\phi_{X}})\hat{x}+(E_{Y}exp^{-i\phi_{Y}})\hat{y}$ (where $\hat{x}\;and\;\hat{y}$ are the unit vectors in x and y-directions, respectively) are presented in vector notation in Figure 5.

As can be seen from the plots, a radially polarized beam is produced when the polarization direction of the incident E-Field is parallel to the singular straight line (fast axis) of the q-plate, transforming the horizontally polarized beam into a “radially polarized” beam. Similarly, a vertically polarized beam is produced when the polarization direction of the incident E-Field is perpendicular to the singular straight line of the q-plate, transforming the vertically polarized beam into an “azimuthal vector” beam.

The slight distortion/deviation from the circularity of the patterns observed in the vectoral elaborations of the E-field patterns (Figure 5) was mainly caused by the mismatch in between the q-plate segments for which they do not perfectly coincide with each other in between the two measurement schemes (*E_X_* vs. *E_Y_* detection). This is for the combined effect of the possible alignment uncertainties due to polarizers’ and antennas’ orientations, and the tilting of the q-plate. Moreover, the junction points in between each segment are relatively thick due to the limitations of the printer utilized, which lead to a slight distortion of the electromagnetic field orientations at the intersection points. This can be reduced by using a higher-precision printer and/or modifying the printer by using a smaller nozzle size.

## 5. Conclusions

We designed a q-plate in a segmented geometry exploiting the concept of form birefringence. The device was fabricated using 3D printing technology. The designed form birefringence was analytically studied and experimentally validated by using THz time domain spectroscopy. It consists of space-variant dielectric slabs made of melt-extruded ABS having an azimuthally variable in-plane rotation, to obtain a fixed rate of change of the optical axis. The device enables the realization of vectorial beams based on diversely space-dependent orientation of the ordinary and extraordinary axes in the transverse plane. As a novel approach, the system adapts *λ*/4 waveplates as the main element, allowing us to achieve radial to azimuthal conversion at discrete frequency intervals. The manufactured q-plate was experimentally tested by a polarization-sensitive imaging routine, and the full position-dependent electric field spatial maps were acquired using the signal propagation through the device. The results are shown to be consistent with the design. E-field azimuthal and radial wave front patterns and the corresponding polarization distribution maps are experimentally demonstrated. The proposed device can be further developed to realize vortex beam generation for possible use in accelerator-based FEL-assisted high-field THz sources.

## Figures and Tables

**Figure 1 micromachines-13-00796-f001:**
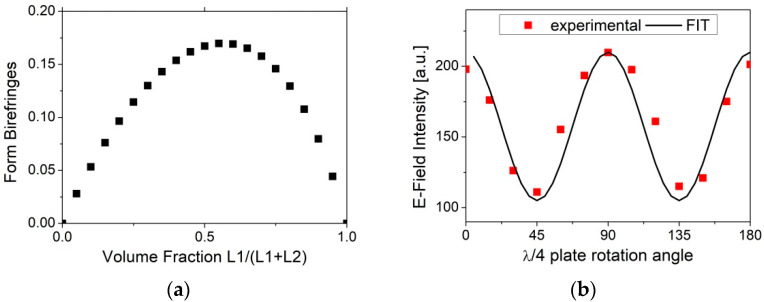
(**a**): Form birefringence vs. dielectric volume fraction calculated using Equations (1) and (2). (**b**): Plot of the E-field intensity of the manufactured *λ*/4 plate device under rotation around the THz beam propagation axis, with 15° steps. The solid curve represents the transmittance for an ideal *λ*/4 plate (see text).

**Figure 2 micromachines-13-00796-f002:**
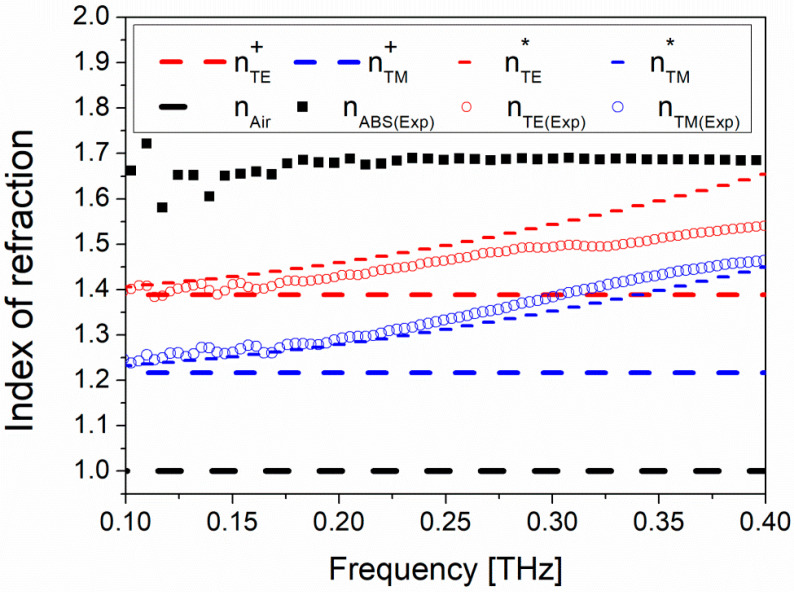
Comparison of the index of refraction calculated from the static case solution ($n_{TE}^{+}$ and $n_{TM}^{+}$, long-dashed curves), the 2nd-order EMT solution ($n_{TE}^{*}$ and $n_{TM}^{*}$, short-dashed curves) and experimentally measured values (open circles), as a function of frequency. The experimentally measured index of refraction for the ABS is also shown using black squares.

**Figure 3 micromachines-13-00796-f003:**
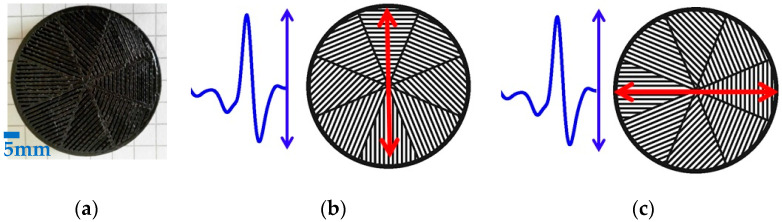
(**a**): Picture of the manufactured q-plate realized using ABS and 3D printing technology. (**b**,**c**): Sketch showing the orientation of the impinging E-field (blue arrow) with respect to the main axis (red arrow) of the q-plate to achieve radial and azimuthal polarization distributions, respectively.

**Figure 4 micromachines-13-00796-f004:**
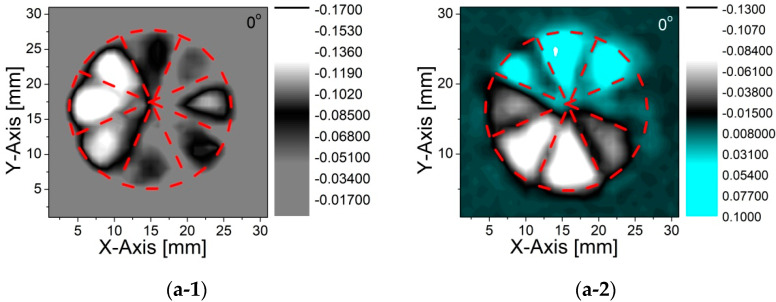
E-Field response of the designed q-plate with respect to the fast axis. The mapped *E_X_* components for 0° (azimuthal counter-clockwise), 90° (radial outwards), 180° (azimuthal clockwise), and 270° (radial inwards) orientations of the device are given with the sequence (**a-1**–**d-1**). Similarly, the corresponding mapped *E_Y_* components are given in sequence (**a-2**–**d-2**).

**Figure 5 micromachines-13-00796-f005:**
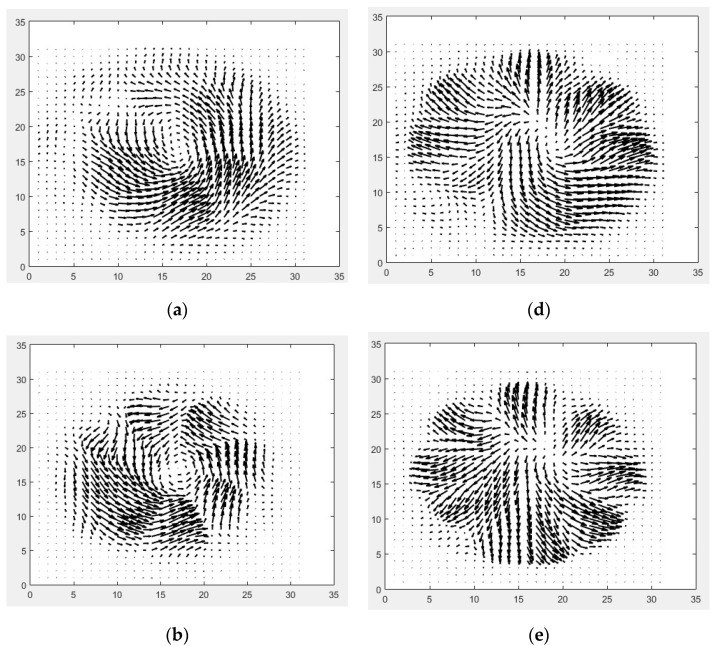
Vector fields corresponding to the E-Field multi-modal response of the q-plate at four main orientations with respect to the fast axis: azimuthal wavefront at 0° (**a**–**c**) and radial wave front at 90° (**d**–**f**) (133, 200 and 266 GHz, respectively).

**Table 1 micromachines-13-00796-t001:** The design frequencies for a multi-mode wave plate.

**m**	**f*_λ_*_/2_ [THz]**	**f*_λ_*_/4_** **[THz]**	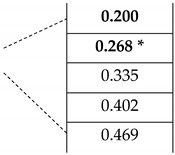
**0**	**0.268 ***	**0.133**	
1	0.804	0.669	
2	1.339	1.205	

## Data Availability

Data underlying the results presented in this paper are not publicly available at this time but may be obtained from the authors upon reasonable request.

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
