# Peer review of "THz Multi-Mode Q-Plate with a Fixed Rate of Change of the Optical Axis Using Form Birefringence"

_micromachines, 2022, doi:10.3390/mi13050796_

Round 1
Reviewer 1 Report
The authors present in this manuscript the experimental work of a THz all-dielectric multi-mode q-plate based on a 3D printing technique. The 3D printing technique can provide a resolution which meets for the requirement for terahertz devices. Although many demonstrations on the use of this technology for terahertz components have been reported to date, this work represents a step towards more complicated devices such as a q-plate. As a result, I recommend the publication of this work after some minor revisions.
- The degree of the form birefringence, as far as one can see from Fig. 1, is pretty small, which is probably due to the low index contrast between polymer/air. If the polymer can be replaced by e.g. silicon, will the form birefringence and the final performance of the q-plate be improved? Actually similar structure based on the use of silicon is not hard to fabrication. One can use the polymer made from 3D printing as the mask for the reactive ion etching of silicon. Some discussions on this point should be added.
- The results presented in Fig. 5 are not fully circled, especially in d, e and f. Some explanations should be provided.
Author Response
REVIEWER 1
“The authors present in this manuscript the experimental work of a THz all-dielectric multi-mode q-plate based on a 3D printing technique. The 3D printing technique can provide a resolution which meets for the requirement for terahertz devices. Although many demonstrations on the use of this technology for terahertz components have been reported to date, this work represents a step towards more complicated devices such as a q-plate. As a result, I recommend the publication of this work after some minor revisions.”
Authors’ response:
We greatly appreciate the reviewer’s positive comments and valuable suggestions on our research work. We have addressed the following constructive comments point by point and carefully revised the original manuscript accordingly. The relevant changes in the manuscript are highlighted in yellow.
“The degree of the form birefringence, as far as one can see from Fig. 1, is pretty small, which is probably due to the low index contrast between polymer/air. If the polymer can be replaced by e.g. silicon, will the form birefringence and the final performance of the q-plate be improved?”
Authors’ response:
To our knowledge at the THz frequencies, the birefringence achieved using periodically structured dielectric/air interfaces are comparable with levels attained by commercially available liquid crystals. We agree with the reviewer that replacing the polymer with silicon would be an alternative to increase the performance of the system. Indeed, with the advent of new technologies in the 3D printing materials/filaments, it is expected to reach higher levels of birefringence, hence increasing the final performance of the q-plate.
To better clarify this point, including relevant citations, we have added in the text the following sentence:
“In the THz region, the birefringence achieved using periodically structured dielectric/air interfaces are comparable with levels attained by commercially available liquid crystals [27]. While replacing polymer with silicon would be an alternative to increase the performance of the q-plate, and also with the advent of new technologies in the 3D printing materials/filaments such as carbon/graphene/MWCNT filler based filaments [21,28] it is expected to reach higher levels of birefringence.”
“Actually similar structure based on the use of silicon is not hard to fabrication. One can use the polymer made from 3D printing as the mask for the reactive ion etching of silicon. Some discussions on this point should be added.”
Authors’ response:
We agree with the reviewer comment. There have been many successful demonstrations in the literature, attaining control over the polarization states of the electromagnetic field by using micro machined subsystems. To our knowledge, some recent works report the possible use of 3D printed shadow mask on Silicon. We think it would be also feasible to use similar techniques for the reactive ion etching of silicon.
Following this suggestion we have added in the introduction the following sentence:
“Some recent works report the possible use of 3D printed shadow masks on silicon[19] showing the feasibility of applying lithography and ion etching techniques to create micromachined subsystems gaining control over the polarization states of the electromagnetic field in a relatively easier and faster way.”
“The results presented in Fig. 5 are not fully circled, especially in d, e and f. Some explanations should be provided.”
Authors’ response:
We thank the reviewer for the constructive comment. Indeed, the EX and EY polarization directions are attained with separate experimental schemes, where the x-component Ex is acquired having the emitter and detector PCA polarizations in the same direction whereas the y-component Ey is obtained having the detector rotated by 90°, such that possibly the segments do not perfectly overlap in between separate measurement schemes. Moreover, the polarizers rotation degrees adding up all the possible alignment uncertainties in between the two measurement schemes, which in turn creates a slight mismatch in the final vectoral elaborations of the E-field patterns leading to a slight deviation from the circularity of the patterns observed in figure 5. It is also important to note that the segment junctions used in between the birefringent slabs are relatively thick (400µm) due to the printing limits of the printer we have used, which can be eliminated by using a higher precision printer and/or by using a smaller nozzle size.
We have added the text phrase below to further clarify the distortion visualized in the vector field plots corresponding to the E-field multi-modal response of the q-plate and further included the possible solutions to the problem:
“The slight distortion/deviation from the circularity of the patterns observed in the vectoral elaborations of the E-field patterns (figure 5) are mainly caused by the mismatch in between the q-plate segments for which they do not perfectly coincide with each other in between the two measurement schemes (EX vs. EY detection). This is for the combined effect of the possible alignment uncertainties due to polarizers and antennas orientations, and tilting of the q-plate. Moreover, the junction points in between each segment are relatively thick due to the limitations of the printer utilized, which lead to a slight distortion of the electromagnetic field orientations at the intersection points. This can be reduced by using a higher precision printer and/or modifying the printer by using a smaller nozzle size.”
We hope that the present version of our manuscript will fulfill the high scientific level of the journal and will be accepted for publication.
On behalf of all authors, I thank you for your kind support.
Yours Sincerely,
Can Koral, PhD
Corresponding Author
Department of Physics “Ettore Pancini”
University of Naples Federico II, Italy
Reviewer 2 Report
Can Koral et al. report the design, fabrication and experimental validation of a THz all-dielectric multi-mode q-plate, which enables the realization of both radial and azimuthal vector beams. The paper is well organized and designed as well as of high quality, and can be accepted for publication when the following issues are addressed.
(1) At THz frequencies, these commonly used approaches are difficult to apply since wave lengths being far larger than structures. How about using plasmonics to manipulate OAM? Such as Nanomaterials 2020, 10, 229, Nanomaterials 2020, 10, 2436, Nanomaterials 2021, 11, 210.
(2) Fig.5 E should be Fig.5 e
(3) The styles of references should be improved.
(4) The legend of some figures should be improved, such as font size.
Author Response
REVIEWER 2
“Can Koral et al. report the design, fabrication and experimental validation of a THz all-dielectric multi-mode q-plate, which enables the realization of both radial and azimuthal vector beams. The paper is well organized and designed as well as of high quality, and can be accepted for publication when the following issues are addressed.”
Authors’ response:
We greatly appreciate the reviewer’s positive comments and valuable suggestions on our research work. We have addressed the following constructive comments point by point and carefully revised the original manuscript accordingly. The relevant changes in the manuscript are highlighted in yellow.
“At THz frequencies, these commonly used approaches are difficult to apply since wave lengths being far larger than structures. How about using plasmonics to manipulate OAM? Such as Nanomaterials 2020, 10, 229, Nanomaterials 2020, 10, 2436, Nanomaterials 2021, 11, 210.”
We agree with the reviewer’s comment. At THz frequencies, the commonly used approaches are difficult to apply since wavelengths are far larger than the birefringent media structures. One solution to this problem can be achieved by using relatively advanced plasmonic structures. Within the light of the suggested references, we have added the recent studies showing the possibility of actively manipulating the spin and orbital angular momentum. We carefully read those articles and further improved the introduction by quoting them in the revised manuscript [12–14].
We have added in the introduction the following sentence:
“One solution to this problem can be achieved by using relatively advanced plasmonic structures. Some recent studies have shown the possibility of actively manipulating the spin and orbital angular momentums at THz frequencies by using graphene-based hybrid plasmonic waveguide structures [12–14].”
“Fig.5 E should be Fig.5 e”
“The styles of references should be improved”
“The legend of some figures should be improved, such as font size.”
Authors’ response:
All typos have been corrected. Reference styles have been changed accordingly to the Micromachines manuscript template style. We have improved the legend of figures in terms of font size and style for an easier visualisation to the readers We have also double-checked grammar, spelling and further improved English writing in the manuscript.
We hope that the present version of our manuscript will fulfill the high scientific level of the journal and will be accepted for publication.
On behalf of all authors, I thank you for your kind support.
Yours Sincerely,
Can Koral, PhD
Corresponding Author
Department of Physics “Ettore Pancini”
University of Naples Federico II, Italy